# Clarifying the quantum mechanical origin of the covalent chemical bond

Daniel S. Levine[1] & Martin Head-Gordon [1,2✉]

Lowering of the electron kinetic energy (KE) upon initial encounter of radical fragments has long been cited as the primary origin of the covalent chemical bond based on Ruedenberg's pioneering analysis of $H_2^+$ and $H_2$ and presumed generalization to other bonds. This work reports KE changes during the initial encounter corresponding to bond formation for a range of different bonds; the results demand a re-evaluation of the role of the KE. Bonds between heavier elements, such as $H_3C-CH_3$, $F-F$, $H_3C-OH$, $H_3C-SiH_3$, and $F-SiF_3$ behave in the opposite way to $H_2^+$ and $H_2$, with KE often increasing on bringing radical fragments together (though the total energy change is substantially stabilizing). The origin of this difference is Pauli repulsion between the electrons forming the bond and core electrons. These results highlight the fundamental role of constructive quantum interference (or resonance) as the origin of chemical bonding. Differences between the interfering states distinguish one type of bond from another.

[1] Department of Chemistry, Kenneth S. Pitzer Center for Theoretical Chemistry, University of California, Berkeley, CA 94720, USA. [2] Chemical Sciences Division, Lawrence Berkeley National Laboratory, Berkeley, CA 94720, USA. ✉email: mhg@cchem.berkeley.edu

The chemical bond is at the very heart of chemistry, as bond strengths determine most of the enthalpic component of the thermodynamic driving forces for reactions, and control key features of molecular structure and properties. The quantum origins of the forces holding atoms tightly and strongly near each other have been the focus of intense discussion as chemistry moved beyond pre-quantum concepts such as the "hooks and eyes" to Lewis's shared electron pairs[1–4]. In a quantum picture, the chemical bond was originally viewed[1], and is still sometimes discussed and taught, as being electrostatic in origin. This was based on the virial theorem: for a (negative) bond energy $\Delta E$, the electron potential energy (PE or $V$) changes (decreases) by $\Delta V = 2\Delta E$; twice as much as the electron kinetic energy (KE or $T$) increases ($\Delta T = -\Delta E$) in an exact quantum calculation at the equilibrium geometry. Additional support comes from the fact that charge accumulates in the internuclear region of a bond relative to the superposition of free atom densities.

Seminal work by Ruedenberg[5] established for $H_2^+$ and $H_2$ that despite the correctness of the virial theorem, roughly 66% of the binding energy can be associated with constructive quantum interference that lowers the KE. KE lowering occurs via delocalization of the electrons' wavefunction across both centers, which is favorable relative to containment in individual 1s atomic orbitals. This process sets up an imbalance between KE and PE relative to the virial theorem, leading to a secondary effect, orbital contraction, in which the orbitals contract toward the nuclei, lowering PE and raising KE. This effect is most easily seen by optimizing the form of a spherical H 1s function as a function of bond length[6]. For the last nearly 60 years, this KE-lowering paradigm has been used to explain the quantum origin of covalent bonds via extrapolation from $H_2^+$ and $H_2$[5–12]. Recently, work has been done to define a schema to allow this theory to be tested with other molecules[11–16]. However, the very strong assumption that the results for hydrogen are universal to other covalent bonds deserves scrutiny by alternative approaches.

Here we use a stepwise variational energy decomposition analysis (EDA) based on absolutely localized molecular orbitals (ALMOs)[17–19] to show that the generalization is, in fact, not universally true in all paradigms. This EDA was designed not with the purpose of separating KE and V but partitioning the total interaction energy, $\Delta E_{INT}$, into well-defined components along the bond-forming path. As all intermediate states in this scheme used to compute energies are valid, spin-pure wavefunctions, subsequent to the EDA's development we were prompted to investigate the role of KE along the bond-forming coordinate. In the course of developing the ALMO-EDA for chemical bonds[17], a term indicating the energy lowering due to orbital contraction was developed[18]. This method quantitatively recovers the results discussed above for $H_2$, and also revealed orbital contraction to be significant for all bonds between hydrogen and other atoms or groups that were examined. Surprisingly though, orbital contraction contributes almost nothing to the bonds between heavier atoms and groups[18]. This was attributed to the presence of core electron pairs in such cases, which precludes significant orbital contraction due to repulsion between the contracting valence and core electrons. This result begs the question that we take up here: if the critical role of orbital contraction is to restore virial balance by raising KE and lowering PE and this does not apparently occur in bonds between non-hydrogen atoms, then what is the nature of the kinetic and potential energy balance in these systems? We utilize the ALMO-EDA method to demonstrate that the model in which covalent bond formation is driven by KE lowering is not universally true for all covalent bonds.

## Results

**Defining the ALMO-EDA for covalent bonds.** The wavefunctions in ALMO-EDA (after assembling fragments into a singlet system) are mean-field for all electrons except for two orbitals (e.g., initially one from each fragment) which may engage in single bond formation. The final wavefunction, $\Psi_{Final}$, is an unconstrained complete active-space SCF [CAS(2,2)] wavefunction) (also known as 1-pair perfect pairing, and two-configurational SCF)[17]. The ALMO-EDA may also be carried out using a density functional theory (DFT) formalism,[19] but we report wavefunction results because the KE in ab initio theory is rigorously defined and does not rely on an approximate exchange-correlation functional. DFT results (using the Kohn–Sham kinetic energy) are qualitatively the same (see Supplementary Table 2 in the Supplementary Information). We summarize the procedure[17–19] briefly below; further details are provided in the "Methods" section.

The interaction energy is $\Delta E_{INT} = E_{Final} - E_{Frag}$, where the energy of isolated fragments is $E_{Frag}$, and the final energy is $E_{Final}$. Using 3 intermediate energies ($E_{Prep}$, $E_{Cov}$, and $E_{Con}$) that are variationally optimized with successively weaker constraints, $\Delta E_{INT}$ is decomposed in stepwise fashion as:

$$\Delta E_{INT} = \Delta E_{Prep} + \Delta E_{Cov} + \Delta E_{Con} + \Delta E_{PCT}. \quad (1)$$

The first term, $\Delta E_{Prep} = E_{Prep} - E_{Frag}$, describes the change in energy as the isolated fragments are distorted from infinitely separated geometrically and electronically relaxed fragments (with $E_{Frag}$) to the geometry and hybridization of the interacting molecule, yielding $E_{Prep}$. For radical fragments with $n_\alpha > n_\beta$, the alpha density is fixed but the beta density is optimized in the span of the alpha density.

The second term, $\Delta E_{Cov} = E_{Cov} - E_{Prep}$, describes the change in energy associated with constructive quantum interference between the prepared wavefunctions of the two individual fragments, subject to the constraint of fixed fragment orbitals. For 2-center, 2-electron chemical bonds, one forms a Heitler–London or valence-bond wavefunction by singlet spin coupling the two fragment unpaired electrons in their overlapping non-orthogonal fragment orbitals (A and B):

$$\Psi_{Cov}^{2e} = c\left\{\hat{\mathcal{A}}[A\uparrow][B\downarrow] + \hat{\mathcal{A}}[A\downarrow][B\uparrow]\right\}, \quad (2)$$

where $\hat{\mathcal{A}}$ is the antisymmetrizer. For 1-electron chemical bonds, the constructive interference on initial bond formation arises from resonating a single electron between the fixed fragment orbitals of the two centers:

$$\Psi_{Cov}^{1e} = c_A \hat{\mathcal{A}}[A\uparrow][B] + c_B \hat{\mathcal{A}}[A][B\uparrow]. \quad (3)$$

Physically, $\Delta E_{Cov}$ allows the wavefunctions of the two prepared fragments to interact, delocalizing the electron or electrons that will form a bond from one fragment to both, and spin coupling them if 2 (or more) are involved. Since the resonance character of (2) and (3) enables an electron (or two) in fixed fragment orbitals to delocalize, $\Delta E_{Cov}$ is the energy change where KE lowering is anticipated, prior to orbital contraction. $\Delta E_{Cov}$ is so named because it will be significant for covalent bonds (and conversely less or not at all significant for more ionic bonds and charge-shift bonds).

$\Delta E_{Con} = E_{Con} - E_{Prep}$ in (1) is the energy lowering due to orbital contraction. In our approach[18], one empty contraction response orbital is obtained per occupied orbital, as the exact linear response of that orbital to perturbing the nuclear charges. In other words, we determine how the fragment density would respond to, for example, an increase in the nuclear charge (by a contraction toward the nucleus) and based on this response, determine what virtual orbitals are necessary to describe this

density change via a coupled-perturbed SCF calculation. The variational energy stabilization obtained by allowing relaxation between occupied orbitals and these contraction orbitals is identified as the orbital contraction energy. This is the step where KE increases are anticipated for traditional covalent bonds. A discussion of how our approach compares with other methods to characterize contraction[12,20,21] is provided in Supplementary Note 2 of the Supplementary Information. We note that the conclusions drawn below about changes in kinetic energy associated with the covalent bond formation step, $\Delta E_{Cov}$, are necessarily independent of the details of the contraction step, which comes afterwards. However, it is interesting to compare the kinetic energy changes of both processes because in $H_2$ the kinetic energy changes during the covalent bond formation necessitate the changes effected by contraction. We will see how these requirements are starkly different in other molecules, using our approach.

Finally, $\Delta E_{PCT} = E_{Final} - E_{Con}$ in (1) is the energy lowering due to bond polarization (P) and charge-transfer (CT). Polarization, which is important for instance in polar bonds, is defined by the variational energy lowering obtained when only those virtual orbitals which describe the response of the fragment to dipolar and quadrupolar fields are included[22]. CT includes the remainder of stabilization associated with CT (allowing us to reach the final 1-PP/TCSCF/CAS(2,2) wavefunction), and is important for instance in charge-shift bonds[20,21,23,24]. In prior uses of the ALMO-EDA method, polarization and CT are considered separately. For the present analysis, however, we lump them together as energy changes that occur after the covalent step, $\Delta E_{Cov}$, that is of primary interest here. Overall, different types of bonds exhibit different "fingerprints" associated with the relative sizes of the terms defined above (e.g., nonpolar covalent bonds have large values of $\Delta E_{Cov}$ and much more modest values for the other terms).

**Analysis of kinetic energy during bond formation.** With energy changes for each step of bond formation defined, let us specify how to test the role of kinetic energy changes in this process. KE changes can be defined for each term of (1): specifically $\Delta T_{Prep} = T_{Prep} - T_{Frag}$, $\Delta T_{Cov} = T_{Cov} - T_{Prep}$, and so on. We consider $\Delta T_{Cov}$, which involves use of the identical set of orbitals in the prepared fragments and the covalently coupled system to be the best test of whether KE lowering accompanies the covalent step of bond formation or not. These numbers are reported below. However, one can imagine alternative definitions, as discussed in Supplementary Note 3 of the Supplementary Information. For example, one alternative is the KE change relative to optimized isolated fragments: $\Delta T_{Cov\text{-}Frag} = T_{Cov} - T_{Frag} = \Delta T_{Prep} + \Delta T_{Cov}$. As shown in Supplementary Note 3, none of the results reported below change qualitatively when other definitions are considered.

We first verify the approach by investigating the well-studied cases of the 1-electron bond in $H_2^+$ and the 2-electron bond in $H_2$, as a function of H–H distance, as shown in Fig. 1. Cumulative binding energies ($E$ rather than $\Delta E$) relative to isolated, geometrically-relaxed fragments as well as changes in kinetic energy ($\Delta T$) relative to the previous step are defined for each step and are what is reported. Thus the figure shows the differences $\Delta T_{Cov}$, $\Delta T_{Con}$, and $\Delta T_{PCT}$ along with the covalent binding energy, ($E_{Cov}$), the binding after contraction ($E_{CON}$), and the final energy ($E_{PCT}$) ($E_{Prep}$ and $\Delta T_{Prep}$ are identically zero). The KE-stabilization picture is clearly recovered: covalent coupling of the fragments results in the majority of the bond strength and a large decrease in KE. Subsequently, contraction contributes a substantial further energetic stabilization (yielding $E_{Con}$) and significantly increases KE ($\Delta T_{Con}$), while final bond polarization and charge resonance produces only modest changes to $T$ and $E$.

To explore 1- and 2-electron bonds between heavier elements, we first examine the central C–C bond in butane and butane cation (chosen because the unpaired electron in the radical cation is well-localized to the central C–C bond). The covalent bond in butane and butane cation should be very similar to that in $H_2$ and $H_2^+$, respectively. However, as Fig. 2 shows, this is not at all the case for KE at equilibrium in $C_4H_{10}$: after total energy-raising and KE-lowering fragment preparation (geometrically distorting from the isolated geometry and orienting the radical orbitals for bonding), $\Delta E_{Cov}$ (i.e., primarily coupling the two spins, as in $H_2$) accounts for an appreciable fraction of the bond energy, but, at equilibrium, bringing two ethyl radical fragments together leads to a substantial increase, not a decrease of the kinetic energy. Contraction, $\Delta E_{Con}$, as was already noted, plays a vanishingly small role energetically both for total and kinetic energy. At equilibrium, the KE is further increased by polarization/CT effects, as was seen for $H_2$ and $H_2^+$. Butane radical cation, on the other hand, is unbound at the covalent and contraction stages, though stabilized significantly from the prepared fragments, and kinetic energy has increased modestly at equilibrium despite this substantial energy lowering (at the unbound energy minimum associated with the covalent level, the kinetic energy decreased slightly). Polarization and CT effects are required to obtain net binding in this weakly bound system. These contributions substantially increase the kinetic energy and lower the total energy.

The origins of the striking differences between $H_2$ and $C_4H_{10}$ may be due to interactions with electrons pairs outside of the bonding pair (other valence pairs and core electrons). Strong support for this hypothesis comes from the behavior of $\Delta T_{Cov}$ for $C_4H_{10}$, at stretched bond distances. Beyond about 1.85 Å, $\Delta T_{Cov}$ becomes negative, suggesting that its increase at equilibrium is associated with other CH electron pairs and/or core electrons. Evidently at shorter distances, interactions between these closed-shell pairs of one ethyl group and the unpaired spin of the other lead to KE-increasing effects, despite the fact that bringing the two fragments together is favorable from a total energy perspective due to constructive wavefunction interference associated with forming the Heitler–London type wavefunction, (2). Similarly, the butane radical cation is substantially stabilized by bringing the two prepared fragments together even though the KE increases slightly, while at longer bond distances, the KE change is negative.

We can visualize the orbitals which are forming the bond along the relaxed butane dissociation curve, and these are shown in Fig. 3. At the stretched distance shown (2.2 Å), we see that the orbitals of the geometrically prepared fragment are just beginning to significantly overlap, but the cores of the atoms are still well separated. This is the location where we would hypothesize the kinetic energy to be most lowered, and indeed this is the minimum of $\Delta T_{Cov}$. Due to the poor overlap of radical orbitals though, the bond is as yet far from fully formed. As we form the final wavefunction at this geometry, electron density can be seen to be pulled toward the other nucleus, increasing the overlap but without significant KE increases. In contrast, at equilibrium, both the bonding orbitals and the non-bonding cores substantially overlap even at the covalent stage, leading to the increase in KE observed in Fig. 2. While subtle to see in this figure, the final orbitals at equilibrium have slightly increased in overlap, but this also increases the collision with the cores, leading to slightly elevated KE from further Pauli exclusion.

**Kinetic energy in first-row hydrides.** In order to obtain further evidence for this hypothesis we consider the valence isoelectronic series $H_2$, LiH, and $BeH^+$ (see Table 1). We have posited that the

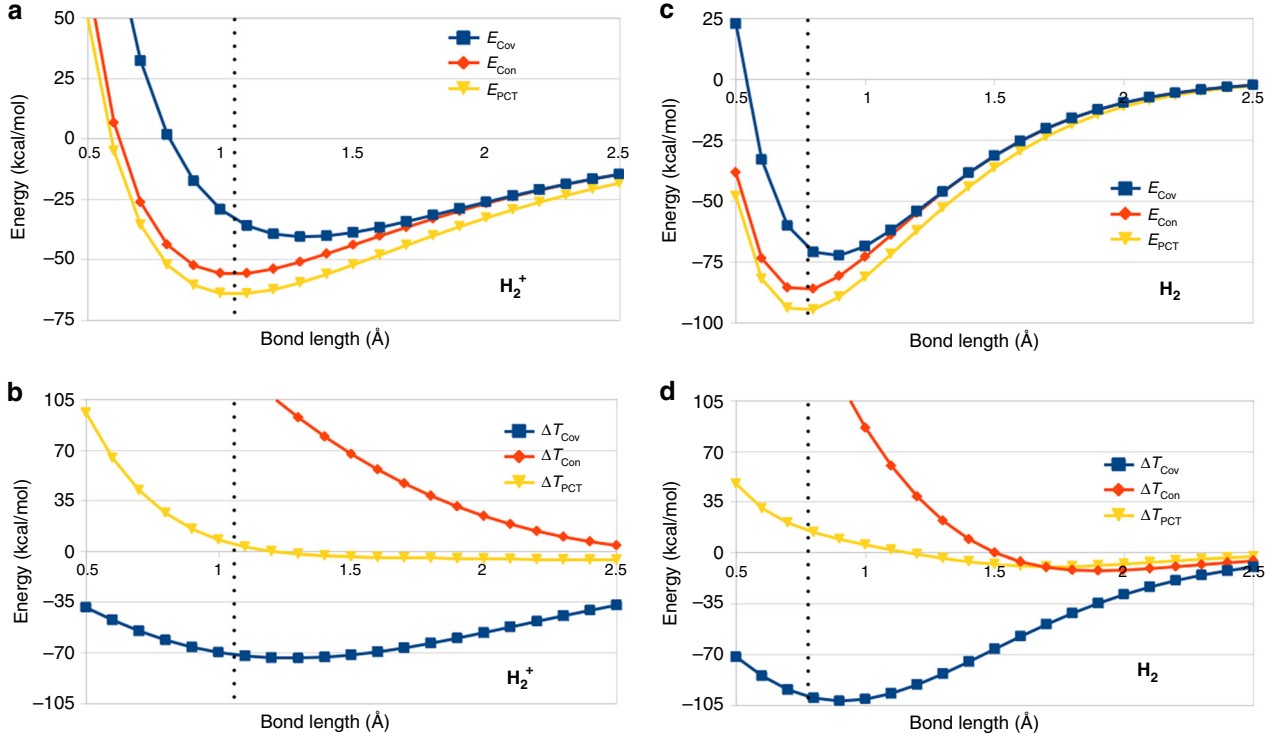

**Fig. 1 Energy and kinetic energy changes for 1 and 2 $e^-$ H–H bonds. a** Energy decomposition (kcal/mol) for dissociation of $H_2^+$; energy terms are cumulative. **b** Kinetic energy decomposition (kcal/mol) for dissociation of $H_2^+$; kinetic energy changes, $\Delta T$, are increments. **c** Energy decomposition (kcal/mol) for dissociation of $H_2$. **d** Kinetic energy decomposition (kcal/mol) for dissociation of $H_2$. Note that a majority of the binding energy occurs due to covalent (Cov) interaction (blue curves in **a**, **c**), accompanied by a significant KE lowering at the equilibrium geometry (vertical dots). The preponderance of the remaining binding energy is contributed by the contraction process, which is accompanied by a substantial increase in KE.

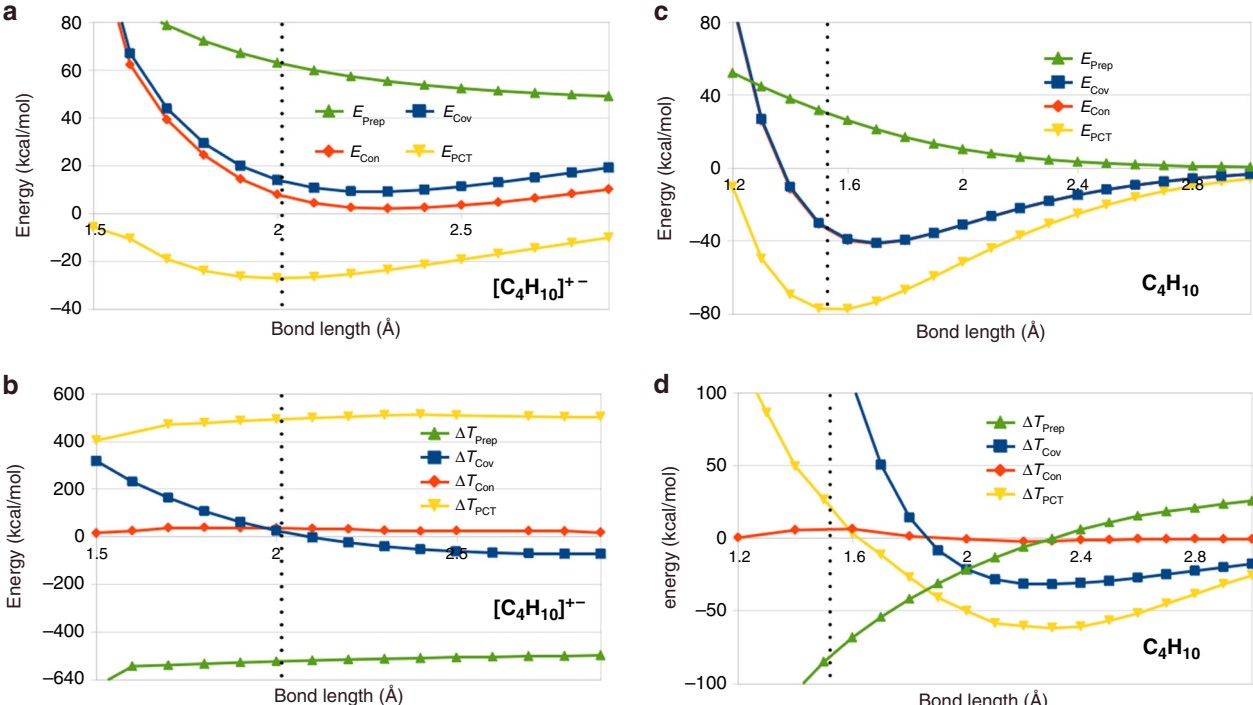

**Fig. 2 Energy and kinetic energy changes for 1 and 2 $e^-$ C–C bonds. a** Energy decomposition for relaxed dissociation of the central C–C bond in *n*-butane cation; energy terms are cumulative. **b** Kinetic energy decomposition for relaxed dissociation of the central C–C bond in *n*-butane cation; kinetic energy changes, $\Delta T$, are increments. **c** Energy decomposition for relaxed dissociation of the central C–C bond in *n*-butane. **d** Kinetic energy decomposition for relaxed dissociation of the central C–C bond in *n*-butane. In *n*-butane, almost half the binding energy occurs due to covalent interaction, but is accompanied by a significant KE increase at the equilibrium geometry (vertical dots). In the weakly bound cation, the KE also increases slightly upon covalent coupling of the fragments at equilibrium. These results are in striking contrast to $H_2^+$ and $H_2$ (see Fig. 1).

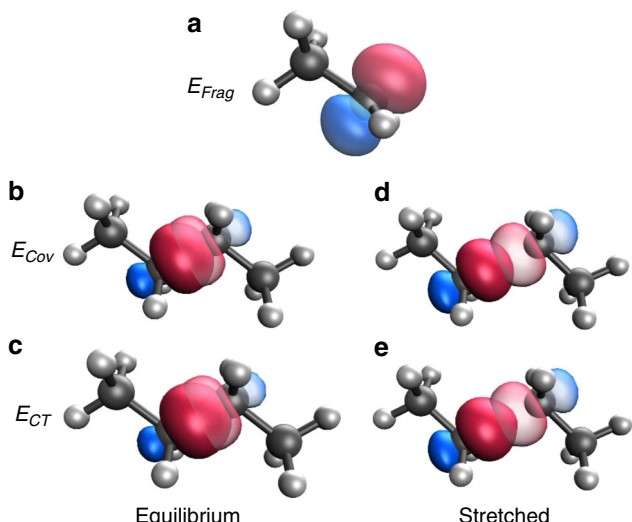

**Fig. 3 Orbitals for C–C bond formation in butane. a** Radical orbital (isovalue 0.1 for all panels) for the ethyl radical at its isolated geometry. **b** Radical orbitals for the nascent C–C bond in the covalent wavefunction at equilibrium (bold for left ethyl, faint for right ethyl). **c** Radical orbitals for the fully optimized wavefunction at equilibrium. **d** Radical orbitals for the covalent wavefunction at stretched (2.2 Å) bond length. **e** Radical orbitals for the fully optimized wavefunction at 2.2 Å.

**Table 1 Energy and kinetic energy stabilizations for A–H bonds (in kcal/mol).**

|  | H–H | Li–H | [Be–H]$^+$ | HBe–H | B–H | C–H |
|---|---|---|---|---|---|---|
| $\Delta E_{Prep}$ | 0.0 | 0.0 | 0.0 | 0.0 | 12.6 | 16.4 |
| $\Delta E_{Cov}$ | −66.0 | −16.8 | −19.2 | −62.7 | −35.4 | −16.9 |
| $\Delta E_{Con}$ | −20.9 | −2.2 | −6.8 | −11.4 | −15.1 | −19.6 |
| $\Delta E_{PCT}$ | −8.4 | −25.5 | −32.4 | −9.1 | −22.9 | −32.1 |
| $\Delta T_{Prep}$ | 0.0 | −0.3 | 0.7 | −4.3 | −15.0 | −17.2 |
| $\Delta T_{Cov}$ | −97.0 | −0.7 | 7.2 | −31.1 | −62.2 | −93.1 |
| $\Delta T_{Con}$ | 182.6 | 38.7 | 76.0 | 94.5 | 113.7 | 124.3 |
| $\Delta T_{PCT}$ | 17.8 | 12.1 | −9.5 | 34.6 | 51.1 | 72.5 |

KE-raising may be due to the overlap of the orbitals principally involved in bonding with core orbitals. For $H_2$, this is impossible (as there are no core electrons) and there is strong lowering of the KE on covalent stabilization. For LiH, the 2s orbital overlaps with the H 1s orbital to form the bond, but there is also overlap of the core Li 1s orbital with the 1s H orbital. The net effect is a dramatically reduced KE lowering (to nearly 0) at the covalent step. In BeH$^+$, the 2s orbital is contracted (due to the positive charge on the Be), resulting in a smaller bond distance (1.3 vs. 1.6 Å) and there is greater overlap of the 1s core of Be with the 1s H electron, resulting in a KE increase at the covalent level. Since orbitals with lower principle quantum number are more rapidly damped by increasing Z than higher ones[25], we would expect that a Be center that had not been contracted by being a cation would have a tighter core and therefore more KE lowering than a Li atom. Indeed, the Be–H bond of BeH$_2$ displays kinetic energy lowering at the covalent stage. On moving to BH and CH, the 1s core is further contracted relative to the 2p orbital. Therefore the B and C 2p orbitals have better overlap with the H orbital, which itself has reduced interaction with the core electrons, enhancing the KE-reduction upon forming these bonds. In all cases, orbital contraction serves to increase KE, though almost all of this effect is due to contraction at the H rather than the core-containing heavy atom. The localization of the contraction was established

**Table 2 Stabilizations for A–H bonds with H contraction orbitals removed (in kcal/mol).**

|  | H–H | Li–H | [Be–H]$^+$ | HBe–H | B–H | C–H |
|---|---|---|---|---|---|---|
| $\Delta E_{Con}$ | −0.3 | −0.1 | −0.1 | −4.3 | −0.4 | −1.2 |
| $\Delta E_{PCT}$ | −29.0 | −27.6 | −39.1 | −16.2 | −37.7 | −50.5 |
| $\Delta T_{Con}$ | 2.2 | −6.5 | 4.4 | 37.8 | 7.7 | −8.1 |
| $\Delta T_{PCT}$ | 198.2 | 57.3 | 62.1 | 91.3 | 157.0 | 204.9 |

by setting the response of the nuclear charge perturbation to zero for hydrogen atoms, thereby making H contraction orbitals unavailable in the contraction orbital optimization step. As shown in Table 2 the result is dramatically reduced energy lowering through contraction on only the heavy atom (cf. Table 1).

**Kinetic analysis of bonds at equilibrium.** Further investigations of a variety of single bond dissociation curves (HF, $F_2$, $Li_2^+$, $Li_2$, Supplementary Figs. 3–10 in the Supplementary Information) and equilibrium positions ($H_3$C–SiH$_3$, $H_3$C–OH, F–SiF$_3$, Supplementary Table 1 in the Supplementary Information) leads to the conclusion that, while single bonds between hydrogen and other atoms or groups lead to large reductions in KE when the groups are brought together without altering the orbitals, heavier groups typically show increases in KE due to the overlap of core electrons. These KE changes are uncorrelated with the total energy stabilization. These data indicate that KE lowering through delocalization is not the universal underlying principle governing the formation of chemical bonds; the wavefunction superposition which lowers the total energy need not be accompanied by a decrease in KE.

**Discussion**
Our results and conclusions may appear controversial in light of the literature discussed so far; therefore some additional comments that connect our work to other existing literature may be helpful. Analysis of electron density deformations from the pro-molecule (superposed atomic densities) to the molecule, going back decades[26,27], shows that orbital contraction is significant for $H_2$ and for HA diatomics but is not significant in AB diatomics. Moreover, these authors discuss that the kinetic energy explanation of the covalent bond as described by Ruedenberg would be insufficient to explain σ-bonds in systems beyond $H_2$, (although they note that π-bonded first-row diatomics might behave similarly to $H_2$ since there are no inner electrons or radial nodes for 2p orbitals) as the involvement of core electrons would necessarily complicate the kinetic energy situation. Early efforts to estimate energy lowering associated with contraction show that, while very significant for hydrogen, it is much smaller for non-hydrogen atoms[28,29] Our work, using very different methods, is broadly consistent with these conclusions concerning contraction, and is the first to show that there is no consistent KE lowering in A–B type bonds, in contrast to H–H and A–H bonds. Shaik, Hiberty, and others, using an entirely different formalism than the one presented here, also have noted apparent kinetic energy increases during bond formation[20,21,23,24]. These authors also ascribe this behavior to Pauli repulsion.

By contrast, recent work by Ruedenberg and co-workers[11–16] indicates that the KE-lowering is critical in all bonds. This qualitative difference is not because of any error on their part or ours. We believe, rather, that it is due to a different choice of reference: our analysis is variational with respect to the radicals that combine to make the bond (i.e., separated atoms and molecular fragments, which are real, well-defined, and quantum mechanical entities). Ruedenberg and co-workers find KE lowering with respect to

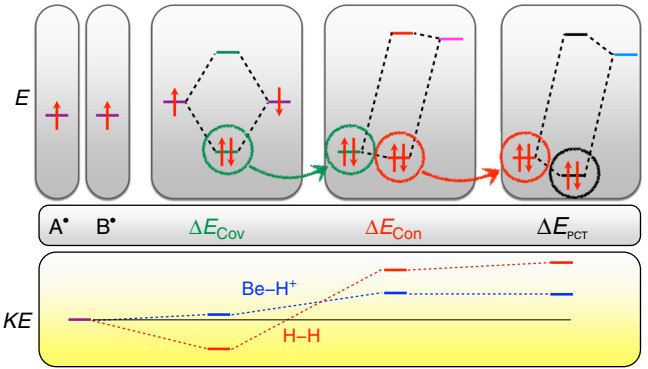

**Fig. 4 Interpretation of the chemical bond.** After orbital preparation, strong interaction between radicals that have degenerate (or close) energy levels gives rise to electron pairing with frozen orbitals ($\Delta E_{Cov}$, in green). This may or may not be accompanied by KE lowering as discussed in the text. Next, further energy lowering occurs by orbital mixing of the initial covalent state (green) with much higher states (pink). Such mixing has the effect of lowering the total energy ($\Delta E_{Con}$, red) and contracting the electron density towards the atomic centers, as shown above the mixing diagrams (exaggerated for visual effect). This effect is significant for H–H and A–H bonds, but much less so for A–B bonds. Mixing with other unoccupied states (light blue) further lowers the energy via polarization and charge transfer ($\Delta E_{PCT}$, black), which is significant for many A–B bonds[19]. While all of these terms are stabilizing in total energy, they may either increase or decrease the KE depending on the system (bottom pane).

artificial quasi-atoms with associated quasi-atomic orbitals that are back-constructed from the final molecular wavefunction. Both the method presented here and the method of Ruedenberg and co-workers are EDAs, descriptive tools for understanding the "why" of bond formation. These chemical concepts are inherently ambiguous and support a variety of reasonable interpretations[30–32]. While many choices for how this can be done are defensible, ultimately a chemical bond is made from interacting atoms and radicals, and should be understood starting from that reference state.

The crux of the matter is schematically illustrated in Fig. 4. Constructive quantum mechanical wavefunction interference leads to stabilization in molecules relative to radical fragments. Wavefunction interference is due to the presence of off-diagonal matrix elements in the Hamiltonian when the fragments are permitted to interact. The Cauchy interlacing theorem (or min–max theorem or Poincare separation theorem or Hylleraas-Undheim-MacDonald theorem) guarantees that, by increasing the size of the Hamiltonian (and adding off-diagonal elements), the eigenvalue spectrum of the Hamiltonian splits, with some eigenvalues going up and some down in energy[33]. This splitting is due to destructive and constructive wavefunction interference, respectively. In particular, what we have shown here is that wavefunction interference with frozen fragment orbitals, $\Delta E_{Cov}$, which lowers the total energy, is significant for all covalent bonds, and this lowering is uncorrelated with changes in the kinetic energy, which may increase or decrease due to the presence or absence of core electrons, in contrast to what has been previously advocated.

The old physical picture does not generalize because it conflated resonance (the true origin of the bond in $H_2$) with delocalization (the physical picture of lowered KE via spreading out the electrons)[34]. The interaction of different electron configurations need not be associated directly with either electron delocalization in space or its anticipated effects on kinetic energy, based on a particle-in-a-box picture. The perspective that wavefunction interference is the origin of the covalent bond has been found by not only this work, but also Ruedenberg, Nascimento, Bacskay, and many others[6,9–15,20,21,23,24,35–44]. The fact that this

picture is recovered by many models speaks to its primacy as the fundamental origin of bonding, while systematic kinetic energy lowering appears in only certain models and in certain systems.

The steps beyond initial covalent bond formation in Eq. (1) are contraction, polarization, and CT. These subsequent orbital optimizations are equivalent to mixing excited configurations into the initial covalent ground state formed with frozen fragment orbitals (see Fig. 4). While this mixing certainly plays a significant role in bond formation, the principal origin of covalent bonding is due to electrons lowering their total energy by superposition with nearly degenerate quantum states that are not fully occupied without mixing in excited states. In one-electron bonds, the nearly degenerate states are due to the electron fluctuating between the bond fragments (e.g., $H^+$–H· and H·–$H^+$). In two-electron bonds, it is the interference between $(\uparrow - \downarrow)$ and $(\downarrow - \uparrow)$ states built from fragment-localized orbitals (in a valence-bond picture) which gives rise to covalent stabilization. Multiple bonds have access to even more states (within a few kcal/mol of the ground state)[45]. It is important to note that for charge-shift bonds like in $F_2$, significant stabilization is also gained by the relaxation of the requirement of fragment locality, which results in a wavefunction (in a generalized valence-bond picture) in which both ionic and covalent structures are present. By contrast, interacting closed-shell fragments do not have low-lying empty quantum states to couple with and so there is no covalent bonding between these moities. One can view this picture as the many-electron analog of the one-electron orbital interaction view of chemical bonding[46]; it is also compatible with valence-bond pictures of chemical bonding[47].

## Methods

**Computational details.** All calculations were carried out with a development version of the Q-Chem program[48]. The aug-cc-pCVTZ basis set was used for all EDA and KE calculations. The geometries used for dissociation curves were those obtained from a constrained HF optimization with the aug-cc-pVTZ basis set. Equilibrium geometries were also obtained from HF/aug-cc-pVTZ calculations.

**Description of ALMO-EDA method.** We here describe the details of the methods used in this work. The method is based on the ALMO-EDA for bonded interactions[17,19].

Step 1: The two halves of the bond are computed as isolated restricted open-shell systems at the geometry that they will adopt in the bonded system. The energy difference (due to geometric distortion) between the infinite separation geometry fragments and the geometry they adopt in the complex is termed $\Delta E_{GEOM}$. In the case of 1-electron bonds, one half of the bond is treated as a radical and the other as a cation (in whichever configuration is lower in energy, all cases studied here are symmetrical).

There is additional electronic preparation which may be included here. Many radicals have a different hybridization than in the corresponding bond. For example, an F atom has an unpaired electron in a $p$ orbital, while an F atom in a bond will be $sp$-hybridized. This preparation may be determined by freezing the $\alpha$ density of the fragment and allowing the $\beta$ density to rotate within that span (recall that the fragments are restricted open-shell) to be optimal in the spin-coupled complex described below. This gives rise to a strictly nonnegative $\Delta E_{Hybrid}$ (in 1-electron bond cases, only the radical fragment has nonzero $\Delta E_{Hybrid}$). As we discuss in detail in ref. [19], even though this is a type of polarization, we include it here because, in some systems, the geometric distortion accounts for some of the electronic preparation (e.g., planar $CH_3$ is $sp^2$-hybridized, while pyramidal $CH_3$ is $sp^3$-hybridized: the electronic preparation cost is tied up in the geometric preparation cost). In either case, including $\Delta E_{Hybrid}$ as part of $\Delta E_{Prep}$ or $\Delta E_{Pol}$, does not change the conclusions of this work. The result of step 1 is a strictly nonnegative preparation energy $\Delta E_{Prep} = \Delta E_{Geom} + \Delta E_{Hybrid}$.

Step 2: These fragments are brought together and assembled into a generally non-orthogonal supersystem in the high-spin triplet configuration by block-diagonally concatenating the molecular orbitals of the fragments. In this way, the orbitals are partitioned ("absolutely localized") to each fragment, giving rise to a "frozen" term $\Delta E_{FRZ}$ which includes electrostatics, Pauli repulsion, and, in DFT, dispersion.

Step 3: A spin-flip is then carried out to form a two-configurational wavefunction, forming a spin-coupled state that employs the fragment orbitals. This energy change $\Delta E_{SC}$ is negative for covalent bonds. It is this spin-coupled system that is relaxed via orbital rotations between the doubly-occupied and half-occupied orbitals to produce the prepared fragments. "Spin-coupling" may be a bit of a misnomer as, in the case of 1-electron bonds (such as $H_2^+$), the SC

wavefunction is the charge-symmetrized wavefunction which contains favorable wavefunction interference but does not actually involve the coupling of two electrons. In this work, $\Delta E_{Cov} = \Delta E_{FRZ} + \Delta E_{SC}$.

Step 4: This wavefunction is then optimized with respect to a set of on-fragment virtual orbitals that describe contraction, $\Delta E_{Con}$. These on-fragment virtual orbitals are those which are necessary to exactly describe the response of the electron density to a perturbation in the nuclear charge[18]. One such contraction orbital is required for each occupied orbital. Depending on how it mixes with its parent orbital, the response orbital may describe either contraction or expansion. In practice, only contraction occurs in the bonding regime.

Step 5: Further on-fragment relaxation is permitted that corresponds to electronic polarization, $\Delta E_{Pol}$. $\Delta E_{Pol}$ is computed by ALMO-constrained optimization of each fragment with a set of on-fragment virtual orbitals. These virtuals are fragment electric response functions (FERFs)[22] that exactly describe the response of the occupied orbitals to uniform electric fields (3 dipolar functions per occupied orbital) and their gradients (5 quadrupolar functions per occupied orbital). The ALMO constraint is a Hilbert space constraint which forbids CT between fragments and the use of the FERF virtual functions provides a well-defined basis set limit for polarization and also ensures that the asymptotic behavior of this term matches the theoretical expectation.

Step 6: Finally, the ALMO constraint is dropped and all orbital rotations are optimized, yielding the CAS(2,2) energy; this final CT energy contribution is termed $\Delta E_{CT}$. In this work, since we do not seek to compare the effects of polarization vs. charge transfer, we aggregate these together as $\Delta E_{PCT} = \Delta E_{Pol} + \Delta E_{CT}$.

For 1-electron bonded systems (all radical cations in this work), the calculation is done as above (with neutral atoms), except that when the ALMO-constrained singlet CSF is formed, the resulting beta electron is removed. This ensures orbital symmetry between the two halves of the 1-electron bonded system.

All intermediates at all steps are valid, spin-pure, properly anti-symmetrized wavefunctions, allowing us to extract kinetic energy, even though this was not the initial goal. The kinetic energy was evaluated at various points during this constrained variational optimization to understand how these different physical processes affect the kinetic energy of the system. We focus on four kinetic energy changes during the bond-forming procedure: the change from relaxed fragments to prepared fragments ($\Delta T_{Prep}$), the change from prepared fragments at infinite separation (which is typically extremely close to that of the electronically relaxed fragments) to the spin-coupled wavefunction ($\Delta T_{Cov}$), the change due to orbital contraction ($\Delta T_{Con}$), and the final change due to polarization and CT ($\Delta T_{PCT}$).

## Data availability

The authors declare that all data supporting the findings of this study are available within the paper and its Supplementary Information files.

## Code availability

All calculations in this work were carried out with a development version of the Q-Chem 5.2 software package. All systems utilized code now available to general users and described in the user manual.

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

## Acknowledgements

This work was supported by grant CHE-1955643 from the U.S. National Science Foundation.

## Author contributions

D.S.L. wrote the code, performed calculations and analysis, and co-wrote the paper. M.H. G. designed the research, performed analysis, and co-wrote the paper.

## Competing interests

The authors declare the following competing interests: M.H.G. is a part-owner of Q-Chem Inc. whose software was used to perform this research; D.S.L. declares no competing interests.
