## [Peer Review File · Nature Communications]

REVIEWER COMMENTS

Reviewer #1 (Remarks to the Author):

Report on Ms. NCOMMS-20-19586-T, "Clarifying the quantum mechanical origin of the chemical bond", by Levine and Head-Gordon

The authors answered most of the points I made in the previous report for Nature Chemistry. This is certainly a very important paper, because it shakes a widely accepted paradigm in chemical bonding. As such, the paper should be accepted after response/correction of points discussed below.

The major and crucial point, which still requires a clear response by the authors, is the orbital contraction issue, and the method used by the authors to gauge contraction:

It is clear that if bringing together two atoms or fragments and letting them form a covalent bond encounters Pauli repulsion (as in $F + F$), then the kinetic energy (KE) increases, and hence the orbital contraction mechanism (which also raise KE) becomes ineffective and has to be replaced by another one. The question is how does one define a unique and invariable "orbital contraction" index/measure? There are 3 methods, which the authors allude to in the paper, and they should be discussed and compared:

(a) The authors of Ref. 26 (and in their recent paper in *ACIE*, 2020, 59, 984), defined orbital contraction as the ratio of the squared of the coefficients of the diffuse parts of the orbitals to those of the contracted parts, and by relating this quantity for the bonded atoms to the same one in the free atoms, one get a clear and intuitive measure of compactness.

(b) Ruedenberg et al (Ref. 12) defined compactness relative to the quasi atoms, and reached the conclusion that compactness is universal. This referee is not quite sure how could the authors of Ref. 12 conclude this, unless the quasi atom undergo some expansion relative to the free atoms.

(c) The authors of the present paper use yet a different criterion for orbital compactness (developed in Ref. 18; see below) and conclude that contraction is limited to bonds with H (e.g., H_2 , $H-A$ $A \neq H$...), while all other $A-B$ bonds ($A, B \neq H$) do not show any compactness upon bonding. While this reviewer agrees with the authors that contraction cannot be universal, still the referee has difficulties to understand the underlying principles of the method used by the present authors. In Ref. 18, they gauge the energetic effect due to orbital-contraction by adding a set virtual response orbitals (one for each occupied orbital), which are optimized in the presence of a monopole (point charge) that is added to the nucleus. While this is surely an ingenious method, it is not intuitively clear without further explanation in the present paper. This would be a pity, because this is the key issue of the paper.

In addition, while the present authors agree with those of Ref. 26 that orbital contraction is not a universal mechanism, they do not agree on the identity of the bonds, which do not show contraction (for example, the C-C bond in Ref. 26 and *ACIE*, 2020, 59, 984 involves orbital contraction, but in the present paper it shows no contraction).

It is extremely important to discuss the issue of contraction and its various definitions (Ref. 12, Ref. 26, and the present paper], so the paper becomes accessible to the entire community, which would be able to compare the method and forms its own view on the role of contraction in chemical bonding.

Relatively minor comments:

(1) The 1-PP/TCSCF/CAS(2,2) wave function for a two-electron bond is not purely covalent and it embeds in the ionic structures through the delocalization tails of the AOs. For all bonds, the

covalent-ionic resonance energies are embedded in the wave function, and for bonds like F-F this resonance is the major effect of bonding. Does this term appear in eq. 1?

(2) On page 5 is written that "CT includes the remainder of stabilization when the ALMO Hilbert space constraint against charge-transfer is removed, and is important for instance in charge-shift bonds." The reader may wonder what are these charge-shift bonds. Why not simply cite some of the appropriate references, e.g., Re 26-28 and ACIE, 2020, 59, 984?

(3) The discussion of H-H on page 5. Here there is a confusing statement "The KE-stabilization picture is clearly recovered: covalent coupling of the fragments results in the majority of the bond strength and a large decrease in KE. Subsequently, contraction contributes a substantial further stabilization that significantly increases KE." How can the contraction be stabilizing and at the same time "significantly increases the KE"? The authors probably mean that the contraction lowers the V and raises T.

(4) The same problem applies to the complex Figure 1, which is not lucid without a clear explanation, of the upper vs. lower Figures for H₂ (or H₂⁺). For example, for H₂, the curve for the total energy (in red) involves contraction and has a deeper curve than the covalent one (blue). But in the Delta-T curve (red) the contracted case rises in energy, etc. May be a clear explanation of Figure 1 would elucidate also Figure 2, which is even more complex.

(5) On page 11, the authors cite Ref. 6, 9-25, 31-40 for studies, which discussed the perspective that the wave function interference is the origin of bonding. The role of constructive interference of the fragment wave functions is also clear in Ref. 26-28 and ACIE, 2020, 59, 984. Why not cite these too?

(6) On page 13, the authors further ascribe the bonding to energy lowering due to the constructive quantum mechanical interference, which is described as a resonance between the two spin arrangements (up/down \leftrightarrow down/up) of the covalent bond. This is not strictly accurate for all bonds. As drawn, in the manuscript, this is a VB description of the Heitler-London wave function with localized AOs. Certainly this is not true for F-F and the like bonds which are described as charge shifted. This is correct however, in the GVB formulation of (up/down \leftrightarrow down/up) where the AOs have tails on the neighboring atoms, and these tails bring about the ionic structures of the bond and stabilize the bond by covalent-ionic resonance. Something of this sort is implied in subsequent lines. It might have been appropriate here to cite Ref. 26-28 and ACIE, 2020, 59, 984, for the valence bond pictures of chemical bonding.

Reviewer #2 (Remarks to the Author):

The authors did an admirable job of refining the manuscript, and preemptively addressed most of the issues I encountered on my third read. At this stage I would defer to the editor for judgement, and I'll include my "remarks to the editor" here in the interests of full disclosure.

(1) The argument for not accepting such a paper is that it is analogous to the "how many angels can dance on the head of a pin" argument for chemists. Different approaches to chemical enlightenment (here, the canonical approach would be that of Ruedenberg, and this approach seeks a Reformation of those beliefs) can be internally consistent, yet contradictory. There is no "right answer," nor can there ever be.

Suggested Revision: I do think it is desirable for the authors to cite some of the perspective/viewpoint articles that discuss the inherent ambiguity of chemical concepts. (A few recent ones: J. Comp. Chem. 40, 2248 (2019); Comp. Theoret. Chem. 1142, 83-87 (2018); Int. J. Quantum Chem. 117, e25359 (2017))

(2) The argument for accepting such a paper is that chemical bonding is the most fundamental

process of chemistry, and having a more nuanced understanding of it is always desirable. Admittedly, this paper is written a bit more forcefully than I would have preferred (it is more a protest against the Ruedenberg picture than an exegesis on bonding from different perspectives), but the critical reader, if nothing else, will realize that there is subtlety in bonding that cannot be grokked from any single perspective (not even the authors').

Summary:

There is nothing wrong with this paper, certainly. At a technical level, it seems perfect. There is nothing "right" either, insofar as the conclusions about chemical bonding are right only within the framework proposed. But these underlying assumptions are clear, and stated with appropriate modesty, so I certainly have no objection to seeing the paper published here.

Reviewer #1:

The authors answered most of the points I made in the previous report for Nature Chemistry. This is certainly a very important paper, because it shakes a widely accepted paradigm in chemical bonding. As such, the paper should be accepted after response/correction of points discussed below.

The major and crucial point, which still requires a clear response by the authors, is the orbital contraction issue, and the method used by the authors to gauge contraction: It is clear that if bringing together two atoms or fragments and letting them form a covalent bond encounters Pauli repulsion (as in $F + F$), then the kinetic energy (KE) increases, and hence the orbital contraction mechanism (which also raise KE) becomes ineffective and has to be replaced by another one. The question is how does one define a unique and invariable “orbital contraction” index/measure? There are 3 methods, which the authors allude to in the paper, and they should be discussed and compared:

(a) The authors of Ref. 26 (and in their recent paper in ACIE, 2020, 59, 984), defined orbital contraction as the ratio of the squared of the coefficients of the diffuse parts of the orbitals to those of the contracted parts, and by relating this quantity for the bonded atoms to the same one in the free atoms, one get a clear and intuitive measure of compactness.

(b) Ruedenberg et al (Ref. 12) defined compactness relative to the quasi atoms, and reached the conclusion that compactness is universal. This referee is not quite sure how could the authors of Ref. 12 conclude this, unless the quasi atom undergo some expansion relative to the free atoms.

(c) The authors of the present paper use yet a different criterion for orbital compactness (developed in Ref. 18; see below) and conclude that contraction is limited to bonds with H (e.g., H_2 , $H-A$ $A \neq H$...), while all other $A-B$ bonds ($A, B \neq H$) do not show any compactness upon bonding.

While this reviewer agrees with the authors that contraction cannot be universal, still the referee has difficulties to understand the underlying principles of the method used by the present authors. In Ref. 18, they gauge the energetic effect due to orbital-contraction by adding a set virtual response orbitals (one for each occupied orbital), which are optimized in the presence of a monopole (point charge) that is added to the nucleus. While this is surely an ingenious method, it is not intuitively clear without further explanation in the present paper. This would be a pity, because this is the key issue of the paper. In addition, while the present authors agree with those of Ref. 26 that orbital contraction is not a universal mechanism, they do not agree on the identity of the bonds, which do not show contraction (for example, the C-C bond in Ref. 26 and ACIE, 2020, 59, 984 involves orbital contraction, but in the present paper it shows no contraction). It is extremely important to discuss the issue of contraction and its various definitions (Ref. 12, Ref. 26, and the present paper], so the paper becomes accessible to the entire community, which would be able to compare the method and forms its own view on the role of contraction in chemical bonding.

A paragraph comparing and contrasting the various definitions of contraction has been added to the Supporting Information.

Relatively minor comments: (1) The 1-PP/TCSCF/CAS(2,2) wave function for a two-electron bond is not purely covalent and it embeds in the ionic structures through the delocalization tails of the AOs. For all bonds, the covalent-ionic resonance energies are embedded in the wave function, and for bonds like F-F this resonance is the major effect of bonding. Does this term appear in eq. 1?

The 1-PP/TCSCF/CAS(2,2) wave function is the final wave function we obtain and we do not expect or desire it to be purely covalent. In the usual presentation of our EDA method, the charge-transfer term is separated from the polarization term, but here we lump them as one "PCT" term since we don't seek to analyze them separately. This term appears in eq. 1. The covalent-ionic resonance in F2 is captured by our CT term in our previous work and by PCT in this work.

To clarify this, we have added "(allowing us to reach the 1-PP/TCSCF/CAS(2,2) wave function)" to the description of the final state after allowing charge-transfer.

(2) On page 5 is written that "CT includes the remainder of stabilization when the ALMO Hilbert space constraint against charge-transfer is removed, and is important for instance in charge-shift bonds." The reader may wonder what are these charge-shift bonds. Why not simply cite some of the appropriate references, e.g., Re 26-28 and ACIE, 2020, 59, 984?

These citations have been added to this section.

(3) The discussion of H-H on page 5. Here there is a confusing statement "The KE-stabilization picture is clearly recovered: covalent coupling of the fragments results in the majority of the bond strength and a large decrease in KE. Subsequently, contraction contributes a substantial further stabilization that significantly increases KE." How can the contraction be stabilizing and at the same time "significantly increases the KE"? The authors probably mean that the contraction lowers the V and raises T .

Contraction contributes further stabilization because the total energy (E) decreases. KE also significantly increases. Of course this is because V decreases by more than KE increases. Throughout the paper, whenever stabilization is referred to, it is the total energy E which is of interest.

To clarify the text, we have changed this passage to say: "...contraction contributes a substantial further stabilization of E and significantly increases KE".

(4) The same problem applies to the complex Figure 1, which is not lucid without a clear explanation, of the upper vs. lower Figures for H2 (or H2+). For example, for H2, the curve for the total energy (in red) involves contraction and has a deeper curve than the covalent one (blue). But in the Delta-T curve (red) the contracted case rises in energy, etc. May be a clear explanation of Figure 1 would elucidate also Figure 2, which is even more complex.

As we hope we have made clear in this paper, the changes in the total energy are not driven directly by changes in the kinetic energy. The reviewer correctly describes what is depicted in Figure 1: that the total energy goes down from contraction but that kinetic energy increases. This is also discussed in the text.

We have further added it as well to the figure caption: "The preponderance of the remaining binding energy is contributed by the contraction process, which is accompanied by a substantial increase in KE."

(5) On page 11, the authors cite Ref. 6, 9-25, 31-40 for studies, which discussed the perspective that the wave function interference is the origin of bonding. The role of constructive interference of the fragment wave functions is also clear in Ref. 26-28 and ACIE, 2020, 59, 984. Why not cite these too?

We have added these citations here as well.

(6) On page 13, the authors further ascribe the bonding to energy lowering due to the constructive quantum mechanical interference, which is described as a resonance between the two spin arrangements (up/down \Leftrightarrow down/up) of the covalent bond. This is not strictly accurate for all bonds. As drawn, in the manuscript, this is a VB description of the Heitler-London wave function with localized AOs. Certainly this is not true for F-F and the like bonds which are described as charge shifted. This is correct however, in the GVB formulation of (up/down \Leftrightarrow down/up) where the AOs have tails on the neighboring atoms, and these tails bring about the ionic structures of the bond and stabilize the bond by covalent-ionic resonance. Something of this sort is implied in subsequent lines. It might have been appropriate here to cite Ref. 26-28 and ACIE, 2020, 59, 984, for the valence bond pictures of chemical bonding.

We have sought to clarify this by the addition of: "It is important to note that for charge-shift bonds like in F₂, significant stabilization is also gained by the relaxation of the requirement of fragment locality, which results in a wave function (in a generalized valence-bond picture) in which both ionic and covalent structures are present."

Reviewer #2:

The authors did an admirable job of refining the manuscript, and preemptively addressed most of the issues I encountered on my third read. At this stage I would defer to the editor for judgement, and I'll include my "remarks to the editor" here in the interests of full disclosure.

(1) The argument for not accepting such a paper is that it is analogous to the "how many angels can dance on the head of a pin" argument for chemists. Different approaches to chemical enlightenment (here, the canonical approach would be that of Ruedenberg, and this approach seeks a Reformation of those beliefs) can be internally consistent, yet contradictory. There is no "right answer," nor can there ever be. Suggested Revision: I do think it is desirable for the authors to cite some of the perspective/viewpoint articles that discuss the inherent ambiguity of chemical concepts. (A few recent ones: J. Comp. Chem. 40, 2248 (2019); Comp. Theoret. Chem. 1142, 83-87 (2018); Int. J. Quantum Chem. 117, e25359 (2017))

These citations have been added along with the sentence: "These chemical concepts are inherently ambiguous and support a variety of reasonable interpretations"

(2) The argument for accepting such a paper is that chemical bonding is the most fundamen-

tal process of chemistry, and having a more nuanced understanding of it is always desirable. Admittedly, this paper is written a bit more forcefully than I would have preferred (it is more a protest against the Ruedenberg picture than a exegesis on bonding from different perspectives), but the critical reader, if nothing else, will realize that there is subtlety in bonding that cannot be grokked from any single perspective (not even the authors').

We have taken this comment to heart and removed some explicit comparisons to the Ruedenberg picture to soften our manuscript. We have also added a new figure, Figure 3, which shows some pictures of the orbitals during butane bond formation in our method which will hopefully provide further nuance and insight into our approach (and therefore toward the exegesis the reviewer (and the authors) rightly desires).

Summary: There is nothing wrong with this paper, certainly. At a technical level, it seems perfect. There is nothing "right" either, insofar as the conclusions about chemical bonding are right only within the framework proposed. But these underlying assumptions are clear, and stated with appropriate modesty, so I certainly have no objection to seeing the paper published here.

REVIEWERS' COMMENTS:

Reviewer #1 (Remarks to the Author):

This is an improved version, which clarified most of the points raised in the previous report. On page 13, one line above the bottom line, change references 20, 23, 24 to 20, 21, 23, 24.

Other than that the present reviewer agree with the authors that kinetic energy lowering is not the general mechanism for covalent or formally covalent bonding.

Response to reviewer comments:

This is an improved version, which clarified most of the points raised in the previous report. On page 13, one line above the bottom line, change references 20, 23, 24 to 20, 21, 23, 24.

Other than that the present reviewer agree with the authors that kinetic energy lowering is not the general mechanism for covalent or formally covalent bonding.

Reply: We have added the additional citation. We are encouraged that the reviewer agrees with the key result of our study.